# Efficient Large-Scale Point Cloud Geometry Compression

**DOI:** 10.3390/s25051325

**Published:** 2025-02-21

**Authors:** Shiyu Lu, Cheng Han, Huamin Yang

**Affiliations:** 1School of Computer Science and Technology, Changchun University, Changchun 130022, China; 2School of Computer Science and Technology, Changchun University of Science and Technology, Changchun 130022, China

**Keywords:** point cloud geometry compression, cross-attention, efficient generation

## Abstract

Due to the significant bandwidth and memory requirements for transmitting and storing large-scale point clouds, considerable progress has been made in recent years in the field of large-scale point cloud geometry compression. However, challenges remain, including suboptimal compression performance and complex encoding–decoding processes. To address these issues, we propose an efficient large-scale scene point cloud geometry compression algorithm. By analyzing the sparsity of large-scale point clouds and the impact of scale on feature extraction, we design a cross-attention module in the encoder to enhance the extracted features by incorporating positional information. During decoding, we introduce an efficient generation module that improves decoding quality without increasing decoding time. Experiments on three public datasets demonstrate that, compared to the state-of-the-art G-PCC v23, our method achieves an average bitrate reduction of −46.64%, the fastest decoding time, and a minimal network model size of 2.8 M.

## 1. Introduction

In the real world, most space is unoccupied by observed objects. Large-scale scene point clouds effectively reduce data volume while preserving spatial structure information, capturing the properties of large-scale scenes. This makes them well-suited for representing the 3D structure of expansive environments, leading to widespread applications in fields such as autonomous driving [1], robotics [2], and virtual reality [3]. However, in practical applications, point cloud data can contain hundreds of millions of points or even reach TB-level sizes. This not only requires substantial storage space but also presents significant challenges in data processing, management, sharing, and application, thereby raising higher demands for point cloud compression. The Moving Picture Experts Group (MPEG) [4] has proposed two traditional point cloud compression standards: video-based V-PCC and geometry-based G-PCC. V-PCC first converts point cloud sequences into video format and applies traditional video compression techniques, while G-PCC uses an efficient octree structure to eliminate redundant information and achieve compression. However, these traditional methods rely on handcrafted features and are limited in their applicability. With the growing use of deep learning in image and video compression, its application to large-scale scene point cloud compression is also gaining increasing attention.

In recent years, deep learning has made significant progress in point cloud compression. Many studies have focused on dense objects or human point clouds. Refs. [5,6,7,8] proposed a series of lossy geometric compression algorithms for point clouds, achieving high reconstruction quality at low bitrates. Refs. [9,10,11,12] introduced a set of lossless geometric compression algorithms, ensuring low bitrates while maintaining lossless quality. However, these algorithms are less effective for large-scale scene point clouds, where the sparse distribution of points makes compression more challenging. This sparsity results in numerous empty voxels during the voxelization of sparse point clouds, significantly hindering subsequent processing. Currently, voxel-based methods for large-scale scene point clouds [13,14,15,16] convert them into octree structures to reduce computational and parameter requirements. However, these methods still demand substantial computation, limiting their applicability. With the advent of efficient models like PointNet [17] and PointNet++ [18], it has become feasible to process point cloud data directly and efficiently. Thus, our approach to the geometric compression of large-scale scene point clouds leverages direct point-based processing. However, due to the high complexity and large scale of such point clouds, existing methods suffer from suboptimal compression performance and complex decoding processes [19,20]. Recent studies [21] have attempted to address these issues by designing an attention-based encoder to enhance compression performance for sparse point clouds and by developing a folding-based point cloud generation module to reduce decoding time. However, the attention module in these designs fails to fully leverage the positional information introduced and does not effectively recover high-dimensional geometric features, resulting in subpar decoding quality. To overcome this, we designed a cross-attention module and an optimized generation module, achieving both lower decoding time and improved decoding quality.

In summary, the main innovations of this work are as follows:We propose an efficient large-scale scene point cloud geometry compression algorithm with a network model size of only 2.8 M, making it suitable for mobile applications.We design a cross-attention module that deeply integrates positional and feature information, enhancing feature extraction quality and improving compression performance.We develop an optimized generation module that effectively recovers high-dimensional geometric features, enhancing decoding quality without increasing decoding time.Extensive comparative and ablation experiments demonstrate that the proposed method achieves state-of-the-art performance across three datasets and delivers superior results in terms of subjective quality as well.

## 2. Related Work

In recent years, substantial progress has been made in large-scale scene point cloud compression, driving rapid advancements in the field. Broadly speaking, existing research can be categorized into two main types: lossy and lossless compression for large-scale scene point clouds. The following is a review of relevant work on geometric information compression for large-scale scene point clouds.

### 2.1. Lossy Geometry Compression for Large-Scale Point Clouds

For lossy large-scale scene point cloud compression, Huang et al. [19] proposed a deep learning-based point cloud compression network that effectively processes various types of point clouds by learning common structural features, including simple and sparse shapes. Wiesmann et al. [20] introduced an innovative convolutional autoencoder architecture that operates directly on the points themselves, thus avoiding voxelization; however, the reconstruction quality remains suboptimal. Liang et al. [22] employed a Transformer-based encoder architecture for point cloud geometry compression, where the input point cloud is treated as a continuous spatial set with learnable positional embeddings and compressed using self-attention layers and point-wise operations. Pang et al. [23] used multilayer perceptrons and convolutional neural networks as the backbone, giving their approach an advantage in handling sparse point clouds and enabling post-processing techniques for further refinement of the decompressed point clouds. Later, Pang et al. [24] proposed a geometry compression algorithm for point clouds that supports point, voxel, and tree representations, featuring a context-aware up-sampling module for decoding and an enhanced voxel Transformer module for feature aggregation. Cui et al. [14] used non-overlapping context windows to construct sequences and shared the results of the multi-head self-attention mechanism, reducing time overhead.

For large-scale point cloud scenes, Sun et al. [25] converted discrete voxels into fine-grained 3D points, effectively addressing the coordinate loss that occurs during the octree generation process. Fan et al. [26] explored the spatial correlation across different layers through progressive down-sampling, modeling the corresponding residuals with a fully decomposed entropy model, thereby achieving compression of latent variables. Inspired by IPDAE [7], Huang et al. [27] proposed an ordered segmentation algorithm based on patch-wise point cloud compression, tailored to the specific characteristics of point clouds. Wang et al. [28] decoupled the original point cloud into multiple layers of point subsets, compressing and transmitting each layer independently to ensure reconstruction quality requirements across different scenarios. You et al. [21] proposed an attention-based encoder that embeds features from local windows and introduces dilated windows as cross-scale priors to infer the distribution of quantized features in parallel.

### 2.2. Lossless Geometry Compression for Large-Scale Point Clouds

For lossless large-scale scene point cloud compression, Huang et al. [13] utilized the sparsity and structural redundancy between points to reduce bitrate. Biswas et al. [29] modeled the probabilities of octree symbols and associated intensity values by exploiting temporal and spatial correlations in the data. However, this work overlooked neighboring nodes and local geometric features in the current point cloud frame. Que et al. [15] proposed a two-stage deep learning framework that combines the strengths of octree-based and voxel-based methods, using voxel context to compress octree-structured data. Fu et al. [16] first employed an octree representation to reduce spatial redundancy, making it robust to point clouds at different resolutions. They then used a conditional entropy model with a large receptive field to model sibling and ancestor contexts, calculating strong dependencies between neighboring nodes, and applied an attention mechanism to emphasize relevant nodes in the context. This extensive context includes ancestor nodes, thousands of neighboring nodes, and their ancestor nodes, covering nearly the entire octree. However, such heavy global attention computations and autoregressive context are inefficient for practical applications.

Building on OctAttention [16], Song et al. [30] introduced a hierarchical attention structure with linear complexity for context scales, preserving a global receptive field. Additionally, they designed a grouped context structure to address the serial decoding issue caused by autoregression while maintaining compression performance. However, since these studies primarily focus on optimizing network structure and context information, they often overlook fundamental training strategies and efficient context utilization. Song et al. [31] later developed a geometry-aware feature extraction module capable of extracting effective features from large-scale contexts. Lodhi et al. [32] employed sparse 3D convolutions to extract features across various octree scales for lossless compression of point cloud octree representations. Luo et al. [33] transformed sparse point clouds from Cartesian to spherical coordinates, simplifying redundancy reduction for neural networks. By applying the spherical coordinate system to Cartesian-based methods like EHEM [30] and OctAttention [16], they demonstrated its effectiveness.

## 3. Proposed Method

The network architecture of the efficient large-scale scene point cloud geometry compression algorithm is shown in Figure 1. This network consists of an encoder, entropy coding module, decoder, and a loss function. During encoding and decoding, the encoder first extracts features from the input point cloud. The Octree-predlift method from G-PCC is then used to encode and decode the down-sampled key points. The entropy coding module estimates the distribution of the geometric features extracted by the encoder and encodes them into a binary file. Finally, the decoder reconstructs the sparse point cloud from the decoded geometric features and key points.

### 3.1. Encoder Module

The encoder is mainly composed of keypoint sampling, window querying, adaptive alignment, and cross-attention modules, as shown in Figure 2. The sampling and querying methods in the proposed algorithm use aggregation operations commonly applied in point cloud analysis tasks, such as farthest point sampling [34], random point sampling [35], and K-nearest neighbor [36]. The encoder module is described in detail below.

Due to the high computational cost of farthest point sampling, the original point cloud $P\in R^{N\times 3}$ is first randomly down-sampled to obtain a subset with no more than M×16 points. Farthest point sampling is then applied to this subset to obtain $P^{bone}\in R^{M\times 3}$ with M points, where $P^{bone}$ represents the key points. Next, adaptive alignment is performed on overlapping local windows, moving each window to the coordinate origin, and finally, rescaling is conducted based on the key points, as follows:(1)$$d=\frac{1}{\left|P^{bone}\right|}\sum_{p_{i},p_{j}\in P^{bone}}\min\{||p_{i}-p_{j}|{|}_{2}:p_{i}\neq p_{j}\},$$(2)$$P_{i}^{W}=\{\frac{p-p_{i}}{d}:p\in T(p_{i},P,K)\},\forall p_{i}\in P^{bone}.$$
where $P_{i}^{W}$ is the aligned local window, $T(p_{i},P,K)$ denotes the K nearest neighbors of point $p_{i}$ found within the input point cloud P, and d is the scaling factor calculated from $P^{bone}$.

#### 3.1.1. Cross-Attention Module

To effectively extract feature information from the local window $P_{i}^{W}\in R^{K\times 3}$, a cross-attention module was designed, as shown in Figure 3. Positional information is introduced to enhance the features, after which the extracted features are concatenated and restored to (K,C) through a multi-layer perceptron. Finally, high-dimensional feature vector $F_{i}^{geo}\in R^{1\times C}$ is extracted using max-pooling and then undergoes entropy encoding.

Specifically, an embedding operation based on graph convolution is first performed on each point within the local window to generate the feature $F_{i}^{\left(0\right)}\in R^{K\times C}$ that captures local details, as follows:(3)$$F_{i}^{\left(0\right)}[j]=GraphConv(T(p_{j}^{i},P_{i}^{AW},K)),\forall p_{j}^{i}\in P_{i}^{AW}.$$
where $T(p_{j}^{i},P_{i}^{AW},K)$ represents the K nearest neighbors of point $p_{j}$ found within the aligned window $P_{i}^{AW}$, GraphConv is defined as GraphConv(⋅)=MaxPool(MLP(⋅)), and $F_{i}^{\left(0\right)}[j]\in R^{1\times C}$ denotes the j-th feature vector in the feature matrix $F_{i}^{\left(0\right)}\in R^{K\times C}$, corresponding to point $p_{j}^{i}$. Next, a stacked cross-attention module is executed, with the process of the l-th attention block described as follows:(4)$$Pem_{i}^{\left(l\right)}=MLP(position),$$(5)$$Key_{i}^{\left(l\right)}=\sigma (MLP(F_{i}^{\left(l\right)}+Pem_{i}^{\left(l\right)})\times F_{i}^{\left(l\right)}+Pem_{i}^{\left(l\right)}),$$(6)$$Value_{i}^{\left(l\right)}=\sigma (MLP(F_{i}^{\left(l\right)}+Pem_{i}^{\left(l\right)})\times Pem_{i}^{\left(l\right)}+F_{i}^{\left(l\right)}),$$(7)$$F_{i}^{\left(l+1\right)}=MLP(Concat(Key_{i}^{\left(l\right)},Value_{i}^{\left(l\right)})).$$
where σ represents the Softmax operation, Pem denotes the positional encoding multiplier, and Key and Value refer to the feature information extracted through an MLP. Finally, the features $F_{i}^{\left(l\right)}\in R^{K\times C}$ output from the last cross-attention module are aggregated into the geometric feature $F_{i}^{geo}\in R^{1\times C}$ using max-pooling, as follows:(8)$$F_{i}^{geo}=MaxPool(F_{i}^{\left(L\right)}).$$

#### 3.1.2. Dilation Window Down-Sampling Module

Since there is dependency between the target local window and the neighboring area of the down-sampled key points, a dilated window is used as a cross-scale prior before entropy encoding. This approach captures the neighboring relationships within the point cloud in the local window, enabling the model to consider both local detail features and the point cloud distribution information over a wider area. Additionally, to achieve fast arithmetic encoding, the geometric features extracted by the encoder are further compressed using graph convolution and fully connected layers. Specifically, as the key points are encoded by G-PCC, the dilated neighborhood can serve as a cross-scale prior. The dilated window $P_{i}^{DW}$ is defined as follows:(9)$$P_{i}^{DW}=T(p_{i},{\hat{P}}^{bone},K),\forall p_{i}\in {\hat{P}}^{bone}.$$
where $P_{i}^{AW}\in R^{k\times 3}$, T represents the K-nearest neighbors of the down-sampled point $p_{i}$ in the down-sampled ${\hat{P}}^{bone}$, with K set to 8 in the experiment.

Due to the longer symbol sequences and higher-dimensional features to be encoded, the complexity of arithmetic encoding increases. Therefore, a simple fully connected layer is used to perform the compression operation, defined as follows:(10)$$f_{i}^{geo}=Linear(F_{i}^{geo}).$$
where $F_{i}^{geo}\in R^{1\times C}$ and $f_{i}^{geo}\in R^{1\times c}$ are the geometric features, with C=128 and c=16 set in the experiment. After arithmetic decoding, a linear layer is also needed to restore the geometric features back to $F_{i}^{geo}$.

### 3.2. Entropy Module

In the entropy encoding module, uniform quantization is employed, which is replaced with added uniform noise during training, resulting in the quantized geometric features denoted as ${\tilde{f}}^{geo}=Q(f^{geo})$, as follows:(11)$$P_{\theta }({\tilde{f}}^{geo})=\prod_{i=1}^{M}\left(L(\Phi^{i})\times U(-1/2,1/2))({\tilde{f}}_{i}^{geo}\right).$$
where $P_{\tau }$ represents the entropy model parameterized by τ, $L(\Phi_{i})$ refers to the Laplace distribution of the quantized features ${\tilde{f}}^{geo}$, and the parameters $\Phi_{i}=(\mu_{i},\sigma_{i})$ and U(−1/2,1/2) denote a uniform distribution over the interval [−1/2,1/2]. The parameter $\Phi_{i}$ can be estimated from the dilated window using a network that includes a GraphConv layer and an MLP, as follows:(12)$$\Phi_{i}=(\mu_{i},\sigma_{i})=MLP(GraphConv(P_{i}^{DW})).$$

Finally, the bit rate of the geometric features is calculated as follows:(13)$$R^{geo}=-\frac{1}{N}\log_{2}P_{\tau }({\tilde{f}}^{geo}).$$
where N represents the number of points in the input point cloud.

### 3.3. Decoder Module

Due to the significant complexity that multi-scale methods may introduce to the decoder, our approach employs a single-scale strategy to reduce computational demands during decoding. Since the number of geometric features M at the key points is much smaller than the original number of input points N, we first process the geometric features using a feature refinement module. Then, we utilize the designed efficient generation module to generate the position information of the points. Finally, the reconstruction of the point cloud is restored to its original position using a backward alignment operation, as shown in Figure 4.

The feature refinement module consists primarily of a dilated window convolution (DW-Conv) and a linear layer, as shown in Figure 5a. The DW-Conv integrates information from the dilated window, reducing redundant computations in the spatial graph and enabling a graph structure for feature convolution. Specifically, using the provided dilated indices, the features of the points within the corresponding dilated window are grouped and collected. Graph convolution (Graph-Conv) is then applied to aggregate these collected groups, resulting in refined features, as illustrated in Figure 5b.

The geometric features processed by the feature refinement module are transformed into point coordinate information using the designed efficient generation module, as shown in Figure 6. Here, an MLP is used for up-sampling the input features, and a Reshape operation adjusts the output dimensions. The efficient generation module employs a dual-branch structure to extract features. The first branch converts the geometric features from a 1×C-dimensional vector into a $R_{\max}\times D$ grid matrix, which is then randomly sampled to produce R×D. The second branch also converts the geometric features into R×D using Reshape and MLP. Finally, the up-sampled geometric features from both branches are concatenated and passed through an MLP and Reshape operation to generate the point coordinates.

The reverse alignment operation is the inverse process of the adaptive alignment operation used in the encoder. Each reconstructed window ${\hat{P}}_{i}^{AW}$ is moved back to its original position and restored to its original scale to assemble into the complete reconstruction result $\hat{P}$, as follows:(14)$$\hat{P}=\cup_{{\hat{p}}_{i}\in {\hat{P}}^{bone}}\left\{(\hat{p}\times \hat{d})+{\hat{p}}_{i}:\hat{p}\in {\hat{P}}_{i}^{AW}\right\}.$$
where $\hat{d}$ is the scaling factor recalculated by ${\hat{P}}^{bone}$.

### 3.4. Loss Function

The chamfer distance [37] is chosen as the loss function to measure the average distance from each point in one point set to its nearest neighbor in the other point set, as follows:(15)$$D_{CD}(P,\hat{P})={\bar{d}}^{2}(P,\hat{P})+{\bar{d}}^{2}(\hat{P},P).$$
where $D_{CD}$ represents the chamfer distance loss function, P denotes the input point cloud, $\hat{P}$ signifies the decompressed point cloud, and ${\bar{d}}^{2}$ is the average symmetric squared distance from the input point cloud to the nearest neighbors in the decompressed point cloud.

The total loss function $L_{total}$ is constructed by adding $D_{CD}$ and the geometric feature bit rate $R_{geo}$, as follows:(16)$$L_{total}=D_{CD}+\lambda R_{geo}.$$
where the calculation of $R_{geo}$ is shown in (13), where λ is the loss weight used to balance the impact of multi-scale loss, optimizing the parameters of the entire network within the end-to-end training scheme.

## 4. Experimental Results

The large-scale point cloud geometry compression algorithm is trained on the Shap-eNet dataset [38], which contains 35,708 point clouds, each generated by uniformly sampling 8000 points from CAD models. The test dataset includes both indoor and outdoor scenes: the indoor point clouds are sourced from S3DIS [39] and ScanNet [40], while the outdoor point cloud map is derived from KITTI [41], processed according to ref. [20]. Detailed information about the test set is provided in Table 1. Our method is implemented using Python 3.10 and PyTorch 2.0 for network model training. The experimental setup includes the following parameters: the optimizer is Adam [42] with an initial learning rate of 0.0005, a batch size of 1, and the model is trained for 140,000 iterations with a local window size of 128. The bitrate–distortion balance is set to 10^−4^, and G-PCC [4] is used for lossless compression of the down-sampled key points. All experiments were conducted on an AMD Ryzen 7 5800X CPU (AMD, Sunnyvale, CA, USA) and an NVIDIA RTX 2080Ti GPU (NVIDIA, Santa Clara, CA, USA).

### 4.1. Evaluation Indicators

In the image domain, objective metrics such as symmetric nearest neighbor, root mean square error, and peak signal-to-noise ratio (PSNR) are commonly used to assess the quality of decoded images. However, point clouds not only involve errors in color attribute information but also errors in geometric data compression. Therefore, it is necessary to evaluate attribute distortion and geometric distortion separately. To assess the data fidelity of compression algorithms, the geometric distortion levels of decoded point clouds at different bitrates are typically compared, a process referred to as bitrate–distortion optimization. In this context, geometric distortion generally refers to the geometric error between the decompressed point cloud and the original point cloud, while bitrate refers to the number of bits required to encode each input point (bpp).(17)$$\mathrm{bpp}=\frac{S}{N}.$$
where S represents the size of the encoded data file in bits, and N denotes the number of points in the original point cloud.

Geometric distortion in point cloud data is typically measured by the distance between the points in the original point cloud and the points in the decoded point cloud. To calculate the distance between a point in the original point cloud and its corresponding point in the decoded point cloud, the correspondence between the original and decoded point clouds must first be established. Then, the distance deviation between corresponding points is used to compute objective evaluation metrics such as geometric peak signal-to-noise ratio. For both the original and decoded point clouds, distortion is calculated separately. A matching relationship is established for the reference point cloud, and based on the corresponding points, coordinate distortion or color distortion is computed. The larger distortion value is taken as the symmetric distortion, commonly referred to as the D1 metric. To calculate the distance from a point to a plane, surface reconstruction of the point cloud is performed first, and then the point-to-surface distance is used in place of the point-to-point distance for the calculation, which is typically called the D2 metric.

The D1 metric is defined by connecting a point $a_{j}$ in the original point cloud A to a point $b_{i}$ in the decoded point cloud B to determine the error vector E(i,j). The calculation formula is as follows:(18)$$e_{B,A}^{D1}(i)=||E(i,j)|{|}_{2}^{2}.$$
where E(i,j) represents the point-to-point error. Based on the distance from all points (i∈B) in the decoded point cloud B to the corresponding points in the original point cloud, the distance is denoted as $e_{B,A}^{D1}$. $N_{B}$ represents the total number of points in the decoded point cloud B. The D1 metric can thus be expressed as follows:(19)$$\varepsilon_{B,A}^{D1}=\underset{b_{i}\in B}{\max}\left\{e_{B,A}^{D1}(i)\right\}.$$

The D2 metric is calculated by projecting the error vector E(i,j) along the normal vector $N_{j}$ to obtain a new error vector E(i,j). This projection accounts for the distance from a point to the plane. The calculation formula for the point-to-plane error is as follows:(20)$$e_{B,A}^{D2}(i)=||E(i,j)|{|}_{2}^{2}={\left(E(i,j)\cdot N_{j}\right)}^{2}.$$

To assess the distortion, the peak signal-to-noise ratio (PSNR) is used. The geometric PSNR is calculated based on the geometric errors D1 and D2, and the formula for calculating the geometric PSNR is as follows:(21)$$PSNR=10\log_{10}\left(\frac{p^{2}}{\max(e_{B,A}^{D_{x}},e_{A,B}^{D_{x}})}\right).$$
where p represents the geometric peak of the decoded point cloud.

### 4.2. Performance Evaluation

The proposed method is compared with the state-of-the-art traditional method, G-PCCv23, where G-PCCv23(O) serves as the default octree-based encoder–decoder. Additionally, a comparative analysis is performed with deep learning-based methods, including OctAttention, IPDAE, and Pointsoup. To ensure a fair comparison, all deep learning-based methods were retrained on the same dataset as our method, and all test samples were normalized to the coordinate range [0, 1023].

To evaluate distortion in lossy compression, we perform an objective comparison using rate–distortion optimization, deriving D1-PSNR from point-to-point error, with bitrate measured in bits per point. Since the test datasets lack reference normals, which are required for calculating D2 PSNR, comparisons for D2 PSNR are not included. Figure 7 shows the rate–distortion optimization curves of our method on the S3DIS, ScanNet, and KITTI datasets, where the proposed method clearly outperforms others in D1-PSNR across all three datasets. Table 2 presents the bitrate gains in D1-PSNR compared to G-PCCv23(O), OctAttention, IPDAE, and Pointsoup, with negative values indicating bitrate savings (i.e., better compression performance). As shown, OctAttention and IPDAE achieve reduced average bitrates across all three datasets compared to G-PCCv23(O), reflecting inferior compression performance. Pointsoup achieves a −44.66% bitrate gain on average in terms of D1-PSNR compared to G-PCCv23(O), while our method achieves a −46.62% bitrate gain, delivering the best experimental results. Table 3 compares the encoding and decoding times of different methods, showing that G-PCCv23(O) has the fastest encoding time, while our method achieves the fastest decoding time. The real-time decoding algorithm we propose is well-suited for tasks such as map construction and updating in dynamic environments. The longer encoding time of our method is attributed to the use of a more complex encoder; however, we plan to design a more efficient encoder module to enable real-time encoding performance. Additionally, as shown in Table 4, our method has the smallest model parameter size of 2.8 M, making it more lightweight compared to G-PCCv2.3 and Pointsoup.

Figure 8, Figure 9 and Figure 10 provide a visual comparison of the point clouds decoded by all methods for both indoor and outdoor scenes. To enhance the visual quality of large-scale point clouds, color rendering is applied, and detailed zoom-in views are provided, indicated by the red and blue boxes in the figures. In Figure 8, the blue zoom-in detail shows that our reconstruction most closely matches the vehicle contours in the original point cloud. The red zoom-in detail highlights that our reconstruction captures the vehicle and human posture details most accurately and completely. In contrast, the Pointsoup method merges the human point cloud with the background, and other methods fail to fully reconstruct the human posture. In Figure 9, the red zoom-in clearly reveals the contours of the chair’s backrest, which are not as distinct in the reconstructions from IPDAE and Pointsoup. The blue zoom-in detail further demonstrates that our method reconstructs the continuous chair legs, whereas other methods exhibit gaps or missing parts. In Figure 10, both the red and blue zoom-in details show that our method successfully reconstructs the chair’s backrest, while the backrest in the IPDAE and Pointsoup reconstructions appears relatively blurry. Additionally, Figure 8, Figure 9 and Figure 10 include bitrate and D1 PSNR comparisons, clearly demonstrating that our method achieves the best reconstruction quality and detail at the lowest bitrate across all tested scene point clouds.

### 4.3. Ablation Experiment

Comparative experiments on different datasets have demonstrated the effectiveness of our algorithm. To further validate the contributions of each module in the proposed network, we conducted ablation studies. First, we performed ablation on the cross-attention module to demonstrate its feature extraction capability. Next, we ablated the efficient generation module to verify its effectiveness in improving point cloud reconstruction quality. During testing, the window size was set to 128, and we compared the bpp and D1 PSNR results of different ablation models, as shown in Table 5. From Table 5, it is evident that both the proposed cross-attention and optimization generation modules are effective for sparse point cloud geometry compression.

For a more intuitive presentation of the ablation study results, we visualized them in Figure 11. When only the proposed optimization generation module is used, it enhances reconstruction quality while reducing the bitrate, as indicated by the brown line in Figure 11. When only the positional attention module is applied, it significantly improves reconstruction quality with a slight increase in bitrate, as shown by the green line in Figure 11. When both modules are incorporated, the network achieves substantial improvements in reconstruction quality while also reducing bitrate, as represented by the red line in Figure 11. In summary, these results strongly demonstrate the effectiveness of each module in the proposed algorithm.

We also compared the encoding and decoding times for different ablation models, as shown in Table 6. The results indicate no significant changes in encoding or decoding times due to module modifications, suggesting that the improved compression performance of the proposed method does not come at the cost of increased computation or parameter size.

## 5. Discussion

The proposed large-scale point cloud geometry compression algorithm addresses the issue of long terminal decoding times by employing an efficient point cloud generation module. However, the encoding time for the method remains relatively long, and future research will focus on optimizing this aspect. Comparative analysis of point-based large-scale point cloud geometry compression algorithms indicates that, while the proposed algorithm outperforms the latest methods, the PSNR for bpp values within the 0–1 range does not exceed 63 dB across the three datasets. This suggests that there is substantial room for improvement in large-scale point cloud geometry compression. Future work will explore more efficient network structures to achieve sparse point cloud geometry compression with higher decoding accuracy while maintaining fast encoding and decoding speeds, promoting broader applications of large-scale point clouds.

## 6. Conclusions

The proposed large-scale scene point cloud geometry compression algorithm achieves state-of-the-art compression performance with the fastest decoding time. In the encoder stage, geometric features are extracted from local windows using a cross-attention module, and dilated windows are introduced as cross-scale priors to enable parallel inference of the quantized feature distribution. These features are then binary-encoded using an entropy coding module. During decoding, the geometric features are first refined, and an efficient optimized generation module is employed to reconstruct the point cloud coordinates. Finally, a reverse alignment operation restores the point cloud in each window to its original scale. Extensive comparative and ablation experiments demonstrate that our method outperforms benchmark algorithms across three open source datasets. Additionally, the proposed algorithm’s neural model size is only 2.8 MB, providing valuable insights for the deployment of large-scale point cloud encoding and decoding technology on mobile devices.

## Figures and Tables

**Figure 1 sensors-25-01325-f001:**
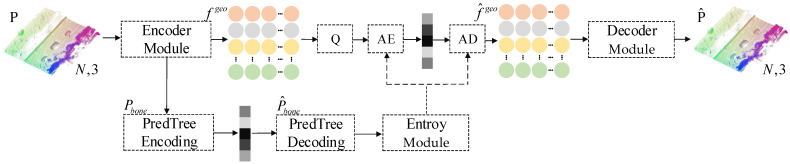
Network architecture overview.

**Figure 2 sensors-25-01325-f002:**
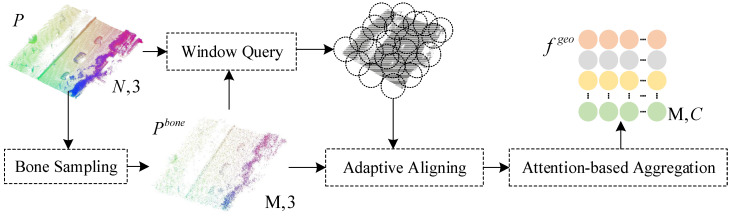
Encoder module.

**Figure 3 sensors-25-01325-f003:**
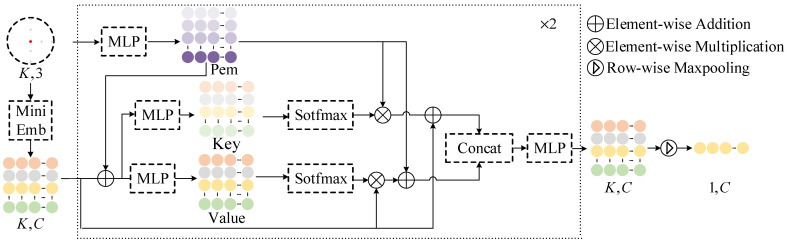
Cross-attention module.

**Figure 4 sensors-25-01325-f004:**
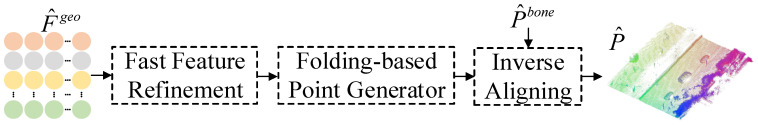
Decoder module.

**Figure 5 sensors-25-01325-f005:**
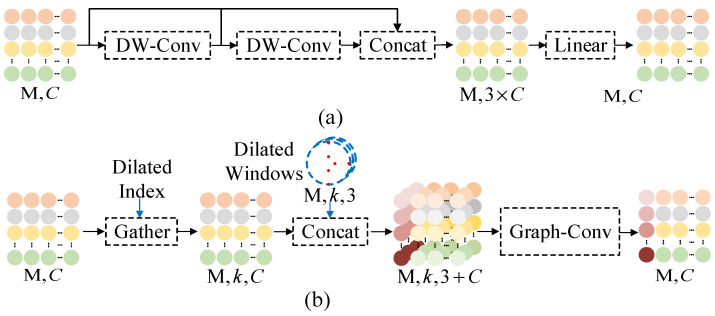
Feature refinement module: (**a**) feature refinement; (**b**) dilated window convolution.

**Figure 6 sensors-25-01325-f006:**
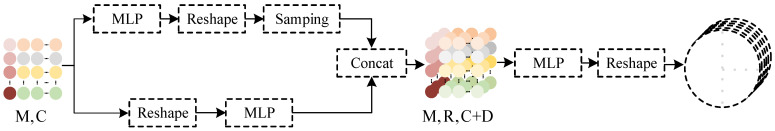
Efficient generation module.

**Figure 7 sensors-25-01325-f007:**
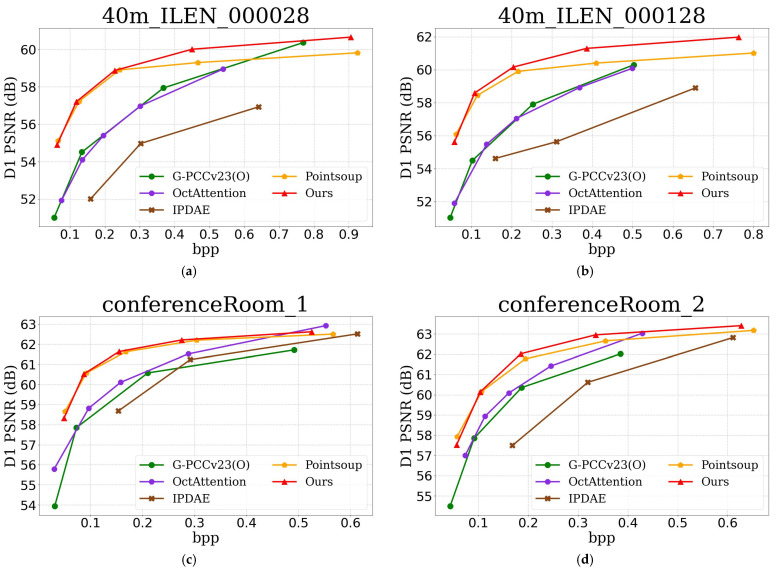
Comparison of bitrate–distortion optimization curves of different methods. (**a**) is the D1 curve of the 40 m_ILEN_000028 point cloud in S3DIS. (**b**) is the D1 curve of the 40 m_ILEN_000128 point cloud in S3DIS. (**c**) is the D1 curve of the conferenceRoom_1 point cloud in ScanNet. (**d**) is the D1 curve of the conferenceRoom_2 point cloud in ScanNet. (**e**) is the D1 curve of the scene0011_00_vh_clean_2 point cloud in KITTI. (**f**) is the D1 curve of the scene0015_00_vh_clean_2 point cloud in KITTI.

**Figure 8 sensors-25-01325-f008:**
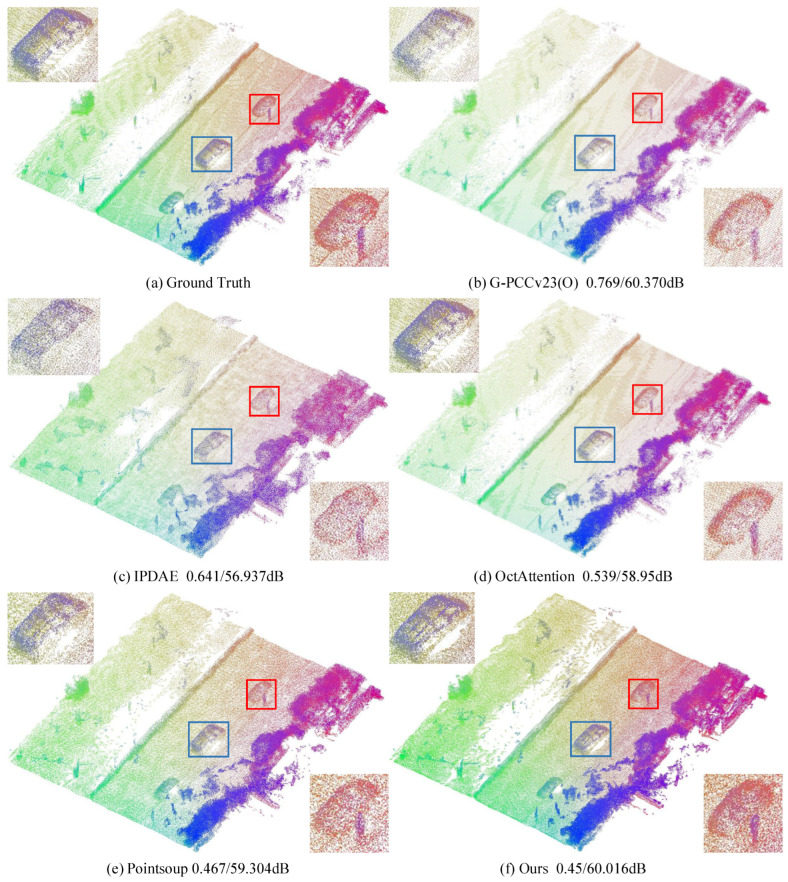
Visual comparison of KITTI dataset. The red and blue boxes represent the magnified details of the decompressed point cloud.

**Figure 9 sensors-25-01325-f009:**
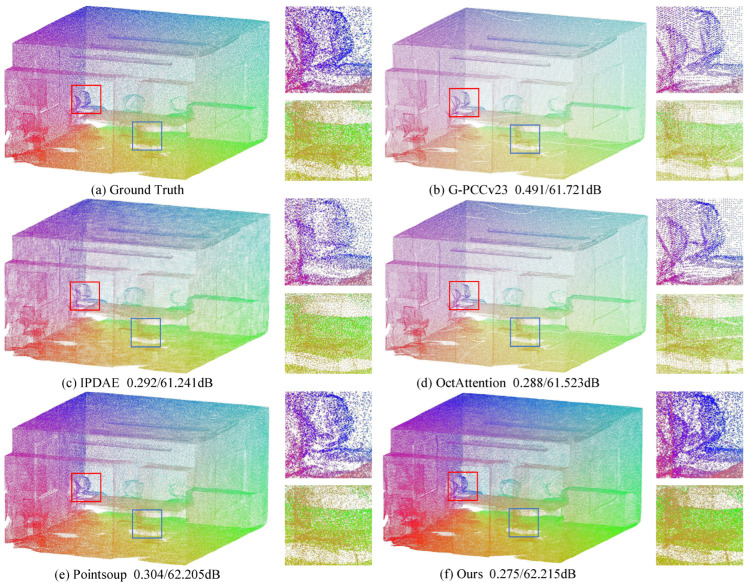
Visual comparison of S3DIS datasets. The red and blue boxes represent the magnified details of the decompressed point cloud.

**Figure 10 sensors-25-01325-f010:**
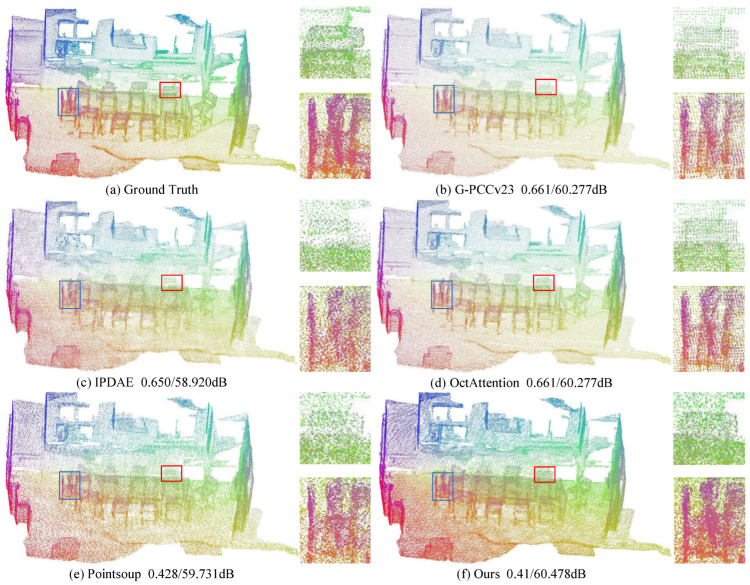
Visual comparison of ScanNet datasets. The red and blue boxes represent the magnified details of the decompressed point cloud.

**Figure 11 sensors-25-01325-f011:**
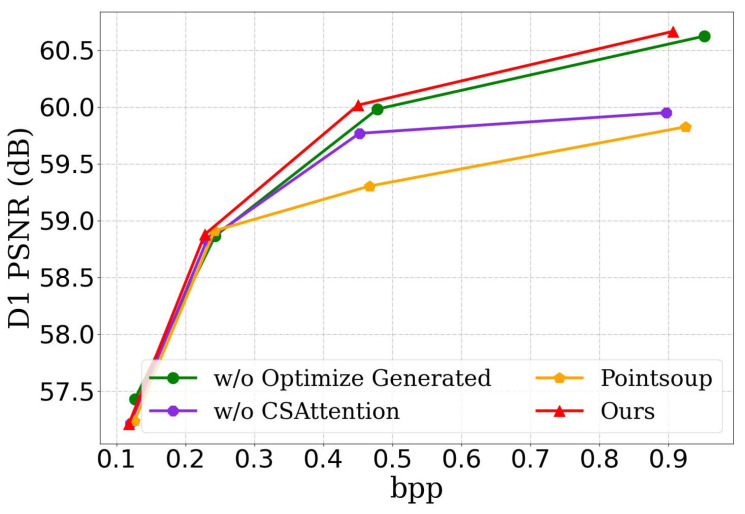
Comparison of module ablation bitrate–distortion curves.

**Table 1 sensors-25-01325-t001:** Test dataset details.

Data	Test	Model Number	Point Number
KITTI	Area 6	48	554 K~214 K
S3DIS	Official test set	100	3.2 M~0.3 M
ScanNet	Sequence 08	186	553 K~32 K

**Table 2 sensors-25-01325-t002:** Bit rate gain of different methods compared with G-PCCv23(O) on D1 PSNR.

Dataset	OctAttention	IPDAE	Pointsoup	Ours
S3DIS	−13.90%	+15.29%	−51.07%	**−51.16%**
ScanNet	+28.34%	+80.80%	−33.35%	**−34.58%**
KITTI	+11.6%	+108.23%	−49.57%	**−54.12%**
Avg.	+8.68%	+68.11%	−44.66%	**−46.62%**

**Table 3 sensors-25-01325-t003:** Comparison of encoding and decoding time of different methods.

	G-PCCv23(O)	OctAttention	IPDAE	Pointsoup	Ours
Times/s	Enc/Dec	Enc/Dec	Enc/Dec	Enc/Dec	Enc/Dec
S3DIS	**0.33**/0.13	0.55/386.60	20.85/1.22	6.96/**0.06**	7.16/**0.06**
ScanNet	**0.06**/0.03	0.15/59.50	4.64/0.24	1.77/**0.02**	1.74/**0.02**
KITTI	**0.11**/0.05	0.17/62.81	6.94/0.46	1.91/0.04	1.97/**0.03**
Avg.	**0.17**/0.07	0.29/169.64	10.81/0.64	3.55/**0.04**	3.62/**0.04**

**Table 4 sensors-25-01325-t004:** Comparison of model size of different methods.

	G-PCCv23(O)	OctAttention	IPDAE	Pointsoup	Ours
Model Size	5.3 MB	28.0 MB	18.8 MB	2.9 MB	**2.8 MB**

**Table 5 sensors-25-01325-t005:** Module ablation comparison.

Cross-Attention Module	Efficient Generation Module	bpp	D1 PSNR
√	×	0.48	59.98 dB
×	√	0.45	59.77 dB
√	√	**0.45**	**60.02 dB**

**Table 6 sensors-25-01325-t006:** Codec time comparison.

	w/o Cross-Attention Module	w/o Efficient Generation Module	Ours
Codec Time/s	1.46/0.04	1.48/0.04	1.49/0.04

## Data Availability

The dataset used in this paper is the publicly available ShapeNat, S3DIS, ScanNet, and KITTI. They can be downloaded at the following links: https://shapenet.org/; https://redivis.com/datasets/9q3m-9w5pa1a2h; https://www.scan-net.org/; https://www.cvlibs.net/datasets/kitti/ (accessed on 31 October 2024).

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
