# Peer review of "Efficient Large-Scale Point Cloud Geometry Compression"

_sensors, 2025, doi:10.3390/s25051325_

Round 1

Reviewer 1 Report

Comments and Suggestions for Authors This manuscript presents a method for large-scale point cloud compression. The experimental shows that this algorithm achieves better results in terms of bit rate gain (Bit rate gain) and encoding and decoding time compared to traditional methods. Here are some suggestions: (1)The manuscript does not explain the calculation methods for PSNR, bpp (bits per pixel), and Bit rate gain. Additionally, the meanings of D1-PSNR and D2-PSNR are not clarified. Please supplement these calculation methods and definitions. (2)Please further elaborate on the details of the visual comparisons in Figure 8, Figure 9, and Figure 10. It is difficult to discern from the visual results that this algorithm shows higher similarity compared to other algorithms when compared to the original algorithm. It is recommended to provide more visual comparisons or quantitative analysis. (3)The encoding and decoding time comparison (Codec Time Comparison) of this algorithm is relatively large. Please explain the specific reasons. Is there potential for optimization? (4)The manuscript lacks further comparison on the smallest model parameter size.   Comments on the Quality of English Language

The discussion section is very short and the language needs to be polished.

Author Response

comments 1:This manuscript presents a method for large-scale point cloud compression. The experimental shows that this algorithm achieves better results in terms of bit rate gain (Bit rate gain) and encoding and decoding time compared to traditional methods.

Response 1: Thank you very much for your comment.

comments 2:The manuscript does not explain the calculation methods for PSNR, bpp (bits per pixel), and Bit rate gain. Additionally, the meanings of D1-PSNR and D2-PSNR are not clarified. Please supplement these calculation methods and definitions.

Response 2:Thank you very much for your comment. The authors sufficiently agree with the comments of the reviewer. We supplement the calculation method and definition of PSNR, bpp and Bit rate gain. The relevant section has been marked in yellow in our revised manuscript. Page 8,9 Line 272-311.

comments 3:Please further elaborate on the details of the visual comparisons in Figure 8, Figure 9, and Figure 10. It is difficult to discern from the visual results that this algorithm shows higher similarity compared to other algorithms when compared to the original algorithm. It is recommended to provide more visual comparisons or quantitative analysis.

Response 3:Thank you very much for your comment. The authors sufficiently agree with the comments of the reviewer. We have provided a more detailed description of Figures 8, 9, and 10. Although the reconstruction results of the proposed algorithm are similar to those of the original algorithm, our method achieves a significantly lower bitrate. Additionally, we have presented bitrate-distortion optimization curves, which provide strong evidence that our proposed algorithm outperforms the original method and other algorithms. The relevant section has been marked in yellow in our revised manuscript. Page 11,12 Line 352-370.

comments 4:The encoding and decoding time comparison (Codec Time Comparison) of this algorithm is relatively large. Please explain the specific reasons. Is there potential for optimization?

Response 4:Thank you very much for your comment. The authors sufficiently agree with the comments of the reviewer. We explain the reasons for the long encoding time and describe the subsequent optimization plan. The relevant section has been marked in yellow in our revised manuscript. Page 10 Line 333-339.

comments 5:The manuscript lacks further comparison on the smallest model parameter size.

Response 5:The manuscript lacks further comparison on the smallest model parameter size. We added a comparison of model sizes for different methods. The relevant section has been marked in yellow in our revised manuscript.Page 10 Line 339-341,344

Reviewer 2 Report

Comments and Suggestions for Authors

The article presents a new method for compressing large-scale point cloud geometry data. It proposes an efficient algorithm leveraging a cross-attention module and an optimized generation module to improve feature extraction, enhance decoding quality, and reduce computational time. The method is benchmarked against existing algorithms and achieves superior compression performance, decoding speed, and model efficiency on three datasets (S3DIS, ScanNet, and KITTI).

The research demonstrates novelty in the design of a cross-attention module for enhancing feature extraction and an efficient generation module for high-quality decoding. These innovations are tailored to sparse, large-scale point clouds, addressing limitations in traditional compression techniques, such as computational complexity and suboptimal reconstruction.

The methodology is rigorous, including comparative analyses, ablation studies, and detailed performance benchmarks on public datasets. The findings are well-supported by experimental results, showing significant improvements in bitrate, decoding speed, and reconstruction quality compared to state-of-the-art methods. The research's clarity and alignment with its objectives are commendable.

Author Response

comments 1:The article presents a new method for compressing large-scale point cloud geometry data. It proposes an efficient algorithm leveraging a cross-attention module and an optimized generation module to improve feature extraction, enhance decoding quality, and reduce computational time. The method is benchmarked against existing algorithms and achieves superior compression performance, decoding speed, and model efficiency on three datasets (S3DIS, ScanNet, and KITTI).

The research demonstrates novelty in the design of a cross-attention module for enhancing feature extraction and an efficient generation module for high-quality decoding. These innovations are tailored to sparse, large-scale point clouds, addressing limitations in traditional compression techniques, such as computational complexity and suboptimal reconstruction.

The methodology is rigorous, including comparative analyses, ablation studies, and detailed performance benchmarks on public datasets. The findings are well-supported by experimental results, showing significant improvements in bitrate, decoding speed, and reconstruction quality compared to state-of-the-art methods. The research's clarity and alignment with its objectives are commendable.

Response 1: Thank you very much for your comments, and I am especially grateful for your recognition of our work.

Reviewer 3 Report

Comments and Suggestions for Authors

The paper aimed at the compression performance and complex encoding-decoding processes to large-scale point clouds, proposed an efficient large-scale scene point cloud geometry compression algorithm, which designed a cross-attention module in the encoder to enhance extracted features and used an efficient generation module to improve decoding quality. And experiments were done  to large-scale point clouds dataset, and the results proved its good performance. However, there are still the following issues in the article that need to be improved by the author:

1In the section 3, Efficient Large-Scale Scene Point Cloud Geometry Compression Algorithm Network Structure is proposed and the subtitles are introduced in detail, but they are not reflected at the Figure 1. It will be better to add the relative modules at Figure 1.

2In Figure 10, the detailed information from the visual comparison in G-PCCv23 is better than the proposed method. Dose the proposed method have some improved way?

3In the Table 3, the proposed method has higher encoding time, which means the whole time is longer. It should be explained its advantages to the some application scenarios.

Author Response

Comments 1:The paper aimed at the compression performance and complex encoding-decoding processes to large-scale point clouds, proposed an efficient large-scale scene point cloud geometry compression algorithm, which designed a cross-attention module in the encoder to enhance extracted features and used an efficient generation module to improve decoding quality. And experiments were done  to large-scale point clouds dataset, and the results proved its good performance.

Response 1:Thank you very much for your comment.

Comments 2:In the section 3, Efficient Large-Scale Scene Point Cloud Geometry Compression Algorithm Network Structure is proposed and the subtitles are introduced in detail, but they are not reflected at the Figure 1. It will be better to add the relative modules at Figure 1.

Response 2:Thank you very much for your comment. The authors sufficiently agree with the comments of the reviewer. We modified Figure 1 and added the relevant modules introduced in Section 3. The relevant section has been marked in yellow in our revised manuscript. Page 4 Line 140-141.

Comments 3:In Figure 10, the detailed information from the visual comparison in G-PCCv23 is better than the proposed method. Dose the proposed method have some improved way?

Response 3:

Thank you very much for your comments. The authors fully agree with the reviewer’s suggestions and have revised the description of Figure 10 accordingly. Since our proposed method achieves a lower bitrate, it further demonstrates the effectiveness of the approach. In the future, we plan to further improve the method presented in this paper by extracting more effective feature information to achieve better reconstruction results. Page 12 Line 364-366.

Comments 4:In the Table 3, the proposed method has higher encoding time, which means the whole time is longer. It should be explained its advantages to the some application scenarios.

Response 4:Thank you very much for your comment. The authors sufficiently agree with the comments of the reviewer. We describe the issues that lead to long encoding times and introduce the advantages of the application scenario. The relevant section has been marked in yellow in our revised manuscript.Page 10 Line 333-337.

Round 2

Reviewer 3 Report

Comments and Suggestions for Authors

As the author revised the manuscript according the issues of interest. The manuscript can be accepted.

Author Response

Thank you very much for your comment.